# TOOLLIBGEN: SCALABLE AUTOMATIC TOOL CREATION AND AGGREGATION FOR LLM REASONING

## ABSTRACT

Large Language Models (LLMs) equipped with external tools have demonstrated enhanced performance on complex reasoning tasks. The widespread adoption of this tool-augmented reasoning is hindered by the scarcity of domain-specific tools. For instance, in domains such as physics question answering, suitable and specialized tools are often missing. Recent work has explored automating tool creation by extracting reusable functions from Chain-of-Thought (CoT) reasoning traces; however, these approaches face a critical scalability bottleneck. As the number of generated tools grows, storing them in an unstructured collection leads to significant retrieval challenges, including an expanding search space and ambiguity between function-related tools. To address this, we propose a systematic approach to automatically refactor an unstructured collection of tools into a structured tool library. Our system first generates discrete, task-specific tools and clusters them into semantically coherent topics. Within each cluster, we introduce a multi-agent framework to consolidate scattered functionalities: a code agent refactors code to extract shared logic and creates versatile, aggregated tools, while a reviewing agent ensures that these aggregated tools maintain the complete functional capabilities of the original set. This process transforms numerous question-specific tools into a smaller set of powerful, aggregated tools without loss of functionality. Experimental results demonstrate that our approach significantly improves tool retrieval accuracy and overall reasoning performance across multiple reasoning tasks. Furthermore, our method shows enhanced scalability compared with baselines as the number of question-specific increases.

## 1 INTRODUCTION

Large Language Models (LLMs) have demonstrated remarkable proficiency in solving complex reasoning tasks (Kojima et al., 2022; Lewkowycz et al., 2022). While Chain-of-Thought (CoT) enhances the reasoning performance by explicitly generating step-by-step natural language processes (Wei et al., 2022), fully using natural language in reasoning reveals fundamental limitations. Firstly, LLMs cannot guarantee high-precision generation, such as complex numerical computation (Gao et al., 2023). Secondly, for procedures that are algorithmic and deterministic in nature (e.g., solving an equation), using a verbose natural language narrative is highly inefficient. This has led to the emergence of authorizing LLMs to invoke external tools, which typically refers to Python functions that can accept specific parameters and output deterministic results (Ma et al., 2024).

The scarcity of tools constrains the widespread adoption of tool-augmented reasoning. For example, in physics question-answering (QA), once the mass and velocity of an object are determined, its momentum is specified by a deterministic relationship. However, a supporting tool for this deterministic relationship is missing. This forces LLMs to rely solely on generating natural language for calculating the momentum, which ideally should be executed by precise programs, thereby introducing unnecessary risks of error and inefficiency. Automated tool-making approaches have been proposed to extract reusable tools from CoT reasoning steps (Qian et al., 2023; Yuan et al., 2023). This process generally involves prompting the LLM with QA datasets and abstracting reusable Python functions as tools from the CoT reasoning outputs. At inference time, a solver LLM first queries the toolset to retrieve the relevant tools and then uses them to answer the question (Ma et al., 2025).

However, a critical bottleneck of the efficacy of the retrieval process emerges in automated tool-making as the set of tools expands. These approaches store question-specific tools into a fragmented

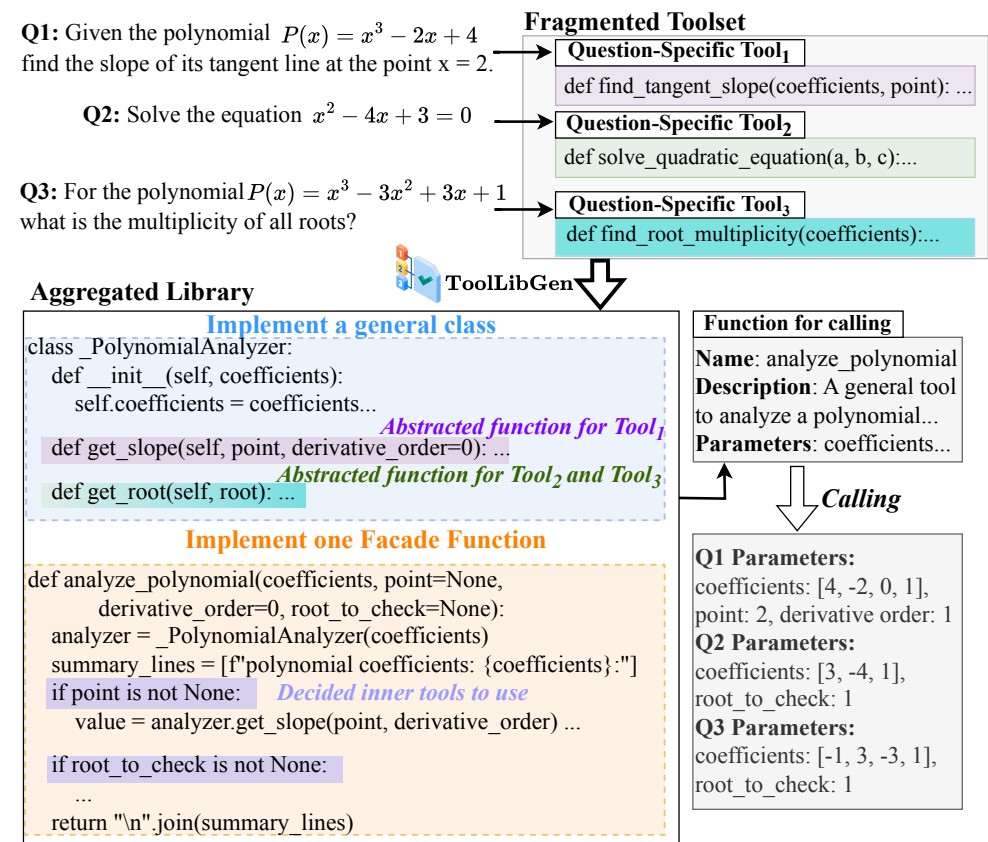

Figure 1: We obtained function-related tools $Tool_{1-3}$ from different problems and $Tool_2$ and $Tool_3$ have overlapping functionality. Through the TOOLLIBGEN, we integrated the three discrete, question-specific tools into a single class that covers all functionalities and a facade function that covers all possible parameter inputs.

tool library, giving rise to two significant retrieval challenges. First, generating question-specific tools for every question results in the linear growth of the toolset and an intractably large search space. Second, the functionally related tools share overlapping capabilities but differ in formulation, causing severe retrieval ambiguity. For example, tools created to solve quadratic equations and those for trigonometric equations might both be named *"get_root"*, but their implementations are vastly different. When trying to solve a problem, a tool with a similar function but not helpful for the current task might be retrieved. Therefore, the bottleneck for scaling automated tool-making has shifted from *"how to create tools"* to *"how to effectively organize those tools"*. In previous research, Didolkar et al. (2024) labeled related tools but did not refactor them, while Ma et al. (2025) merged tools through pairwise comparisons, where each merging step was limited to examining only a small number of tools. These approaches perform only local adjustments and thus have a limited impact.

As shown in Figure 1, we argue that once these tools are generated, a global design should be applied to refactor a large group of functionally related tools into a Python library, comprising standardized classes and public functions, through a higher level of abstraction. It can more significantly reduce the number of tools but keep the same functionality because its goal is not merely to eliminate direct, superficial redundancies, but to abstract and unify the common *core functionality* behind different tools through in-depth analysis.

In this paper, we propose a pipeline, TOOLLIBGEN, to automatically refactor a fragmented collection of ad-hoc tools into a structured library (Figure 2). The first challenge is identifying functionally related tools within a large, fragmented collection. To address tool identification, we employ a hi-

erarchical clustering module that groups tools into coherent sub-domains. The second challenge is aggregating these related tools without losing the specific functional nuances of each implementation, as we observe that a naive, single-pass refactoring by a single LLM frequently overlooks critical details from the original tools. Therefore, the cornerstone of our approach is a multi-agent collaborative framework designed for robust, iterative refinement. TOOLLIBGEN establishes a feedback loop between a *Coding Agent*, which synthesizes a unified implementation for each tool cluster, and a *Reviewing Agent*, which rigorously validates this code for complete functional fidelity against the original tools, and generates targeted feedback for revision. This iterative process continues until the *Reviewing Agent* determines that the aggregated library is sufficiently effective for the solver LLM to answer the tool-making question and fully preserve the capabilities of the initial toolset.

We conducted extensive experiments across diverse reasoning tasks, including science, math, and medical QA, upon multiple foundation models, including GPT-4.1 (OpenAI, 2025a), Qwen3-8B (Yang et al., 2025), and GPT-oss-20B (OpenAI, 2025c), to validate our pipeline. Our structured tool library significantly outperforms baseline methods that use fragmented ad-hoc tool collections. Our approach improves the success rate of seen tasks by an average of over 5%-10%, underscoring the superior retrieval accuracy of the library structure. Moreover, our method demonstrates strong generalization, improving accuracy by over 3% on entirely unseen questions. Furthermore, we provide a detailed analysis showing that the accuracy improvement of our method stems from enhanced retrieval, which is achieved by merging functionally related, question-specific tools.

## 2 TOOLLIBGEN: SCALABLE TOOL LIBRARY GENERATION

### 2.1 OVERVIEW

We establish a systematic methodology for constructing a Python library from a QA dataset. Given a QA dataset $\mathcal{D}$, consisting of $N$ pairs of question $Q$ and its corresponding CoT reasoning trace, denoted as $\mathcal{D} = \{(Q_i, CoT_i)\}_{i=1}^{N}$, our system ultimately generates a Python library, denoted as $\mathcal{L}$, that consolidates all reusable deterministic logic extracted from the CoT traces.

As illustrated in Figure 2, the pipeline unfolds in three core stages. First, in the **Question-Specific Tool Creation** stage, question-specific Python functions are abstracted as tools from each question-CoT pair. After processing all questions in $\mathcal{D}$, we obtain a collection of tools $\mathcal{T} = \{t_1, \ldots, t_M\}$, where $M$ is the number of all question-specific tools. Second, in the **Tool Clustering** stage, the workflow groups $\mathcal{T}$ into multiple clusters $\mathcal{C} = \{C_1, \ldots, C_K\}$, where each cluster consists of multiple tools, denoted as $C_k = \{t_{k,1}, \ldots, t_{k,m_k}\}$, with $m_k \in [1, K]$ and $m_k \ll M$ for all clusters. Third, in the **Tool Aggregation** stage, the tools within each cluster $C_k$ are consolidated into cohesive Python *classes* and auxiliary *functions*, denoted as $C_k^* = \{t_{k,1}^*, \ldots, t_{k,m_k^*}^*\}$. Each aggregated tool $t^*$ is realized as one or more Python *classes* with an associated interface *function*, encapsulating the functionalities of multiple original tools. This significantly reduces the number of tools in $C_k$ and eliminates redundant functionality. Finally, the complete Python library is given by $\mathcal{L} = \{C_k^*\}_{k=1}^{K}$. Our framework employs two LLMs. One general LLM orchestrates the whole pipeline, while another LLM $LLM_{\text{solver}}$ is used in the validation process to use the generated tools to solve questions.

### 2.2 QUESTION-SPECIFIC TOOL CREATION

The initial phase focuses on creating question-specific tools for each question. The LLM first abstracts a reusable toolset from each pair of $(Q_i, CoT_i)$. Each tool $t$ in the toolset is a Python function, including the function signature and implementation. In practice, the LLM can occasionally generate tools that do not meaningfully contribute to reasoning. To mitigate this, we introduce a validation-and-refinement mechanism. Specifically, for each tool $t$, $LLM_{\text{solver}}$ attempts to solve the original question $Q_i$ using the tool. If tool $t$ is used correctly and leads to a correct answer, it means $t$ is helpful in reasoning and is eligible to be accepted into the collected toolset $\mathcal{T}$. Otherwise, the LLM receives the full context of the failure trajectory and generates an improved version of the tool, which replaces the previous version in the candidate set. This iterative loop ensures that each tool in $\mathcal{T}$ helps solve question $Q$. By repeating this process for all questions in the dataset $\mathcal{D}$, the creation module produces multiple question-specific tools, forming the fragmented collection of tools $\mathcal{T}$.

### 2.3 TOOL CLUSTERING

Given the large number of tools in the fragmented set $\mathcal{T}$, it is crucial to determine which tools should be aggregated together before performing aggregation. We achieve this by grouping functionally

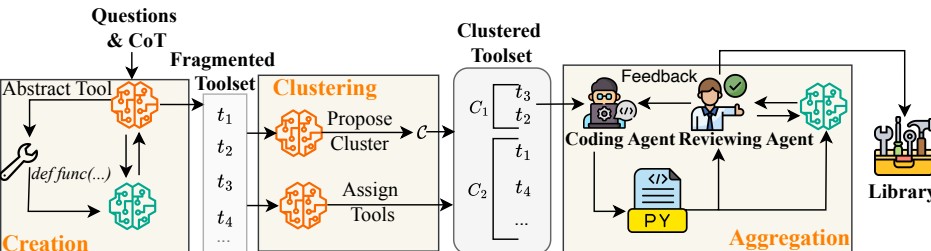

Figure 2: The pipeline of our proposed method (🌐: General LLM; 🌐: $LLM_{solver}$). The LLM first generates and validates question-specific tools. Then it proposes clusters and assigns each tool to specific clusters. For each cluster, a coding agent abstracts the functionality of tools and writes code, while a reviewing agent validates the code with $LLM_{solver}$ to preserve the original functionalities.

related tools into a cluster. We expect that tools within the same cluster are closely related, and argue that grouping should respect academic disciplines, as tools from fundamentally different domains, such as physics and biology, are inherently heterogeneous and should not be combined.

Directly clustering tools into sub-domains is challenging because the full set of domains is unknown. To address this, we first randomly select a small subset of seed tools $\mathcal{T}_{seed} \subset \mathcal{T}$ and prompt an LLM to construct a tree-structured clustering, which encourages the LLM to produce fine-grained clusters, with nodes closer to the root corresponding to broad categories (e.g., physics) and deeper nodes representing increasingly specific subdomains (e.g., kinematics). The tree depth is limited to 4 to prevent excessive subdivision. In practice, we find that the functionalities of tools within the leaf clusters are sufficiently distinct, so it is sufficient to retain only the leaf nodes for subsequent aggregation. The initial tree produces a set of $K$ leaf nodes $\{label_k\}_{k=1}^K$. The remaining tools are then processed iteratively for updating the current tree by creating a new leaf node or merging multiple leaf nodes. After the updating, each tool is assigned to one or more leaf nodes, resulting in the final set of clusters $\mathcal{C} = \{C_1, \ldots, C_K\}$, where each $C_k$ contains $m_k$ tools $(t_{k,1}, \ldots, t_{k,m_k})$.

## 2.4 TOOL AGGREGATION

In this stage, the goal is to consolidate the tools within each cluster $C_k = \{t_{k,j}\}_{j=1}^{m_k}$ into a smaller set of aggregated tools $C_k^* = \{t_{k,j}^*\}_{j=1}^{m_k^*}$, where $m_k^* \ll m_k$. Each aggregated tool $t_{k,j}^*$ encapsulates the functionalities of multiple original tools $t_{k,j}$ and is implemented as one or more Python *classes* with an associated interface *function* that provides a unified entry point. This design allows multiple, functionally related tools to be combined into cohesive, object-oriented abstractions, reducing redundancy, clarifying tool responsibilities, and simplifying downstream usage.

We first have a code agent, which is responsible for the abstraction. Its action space includes designing a plan and writing code. Specifically, the first action is to formulate an overall blueprint based on enumerating all tool names and codes in the current cluster. The blueprint specifies the set of scenarios covered by the current cluster (e.g., numerical analysis) and the mapping from original question-specific tools (e.g., *find_tangent_slope*) to a more general abstraction (e.g., the *Polynomial-Analyzer* class and a facade function), as exemplified in Figure 1. Based on this structured design, the code agent then proceeds to generate the concrete code to implement this blueprint.

However, relying solely on a single round of aggregation remains problematic. In particular, during code generation, the LLM often overlooks a substantial number of extracted tools, leading to missing functionalities. To address this, we incorporate a refinement mechanism that supports iterative, multi-round code improvement. A straightforward option would be to add a reviewing action to the Code Agent's action space, similar to the approach in the Tool Creation module. However, the Code Agent processes all tools in a lengthy context, and additional rounds of generation would further increase context size, potentially degrading performance (Liu et al., 2025). Moreover, reviewing is largely independent and does not require access to the entire context maintained by the Code Agent. For these reasons, we introduce a separate Reviewing Agent, forming a multi-agent system in which the Code Agent and Reviewing Agent collaborate to iteratively enhance overall code quality.

---

**Algorithm 1** Algorithm of TOOLLIBGEN

---

**Input:**
    Dataset $\mathcal{D} = \{(Q_i, CoT_i)\}_{i=1}^{N}$
**Output:** Python library $\mathcal{L}$
1: $\mathcal{T}, \mathcal{L}, \mathcal{C}$, SourceQuestionMap, SourceQuestionMapForCluster $\leftarrow \emptyset, \emptyset, \emptyset, \{\}, \{\}$
                                                             ▷ **Phase 1: Question-specific Tool Creation**
2: **for** each $(Q_i, CoT_i) \in \mathcal{D}$ **do**
3:      $\mathcal{T}_i \leftarrow LLM(\text{“Abstract”}, Q_i, CoT_i)$
4:      **for** each tool $t$ in $\mathcal{T}_i$ **do**
5:          **while** $turn <$ Max_Checking_Turns **do**
6:              Trajectory $\leftarrow LLM_{solver}(\text{“Solve the question”}, Q_i, t)$              ▷ Solve with $LLM_{\text{solver}}$
7:              Decision $\leftarrow LLM(Q_i, CoT_i, t, \text{Trajectory})$
8:              **if** Decision is Pass **then break**
9:              $t \leftarrow LLM(\text{“Refine”}, t, \text{Trajectory})$
10:          $\mathcal{T} \leftarrow \mathcal{T} \cup \{t\}$; SourceQuestionMap$[t] \leftarrow (Q_i, CoT_i)$

                                                                 ▷ **Phase 2: Tool Clustering**
11: $\mathcal{T}_{\text{seed}} \leftarrow$ RandomSample$(\mathcal{T}, m)$; $\mathcal{T}_{\text{rem}} \leftarrow \mathcal{T} \setminus \mathcal{T}_{\text{seed}}$
12: $\{label_k\}_{k=1}^{K} \leftarrow LLM(\text{“Propose Clusters”}, \mathcal{T}_{\text{seed}})$
13: **for** each batch $\mathcal{T}_b$ of size $b$ in $\mathcal{T}_{\text{rem}}$ **do**
14:      $\{label_k\}_{k=1}^{K} \leftarrow LLM(\text{“Update Clusters”}, \{label_k\}_{k=1}^{K}, \mathcal{T}_b)$                 ▷ K may change
15: $\mathcal{C} \leftarrow \{C_k\}_{k=1}^{K}$, each $C_k \leftarrow \emptyset$ from $\{label_k\}_{k=1}^{K}$
16: **for** $t \in \mathcal{T}$ **do**
17:      $k \leftarrow LLM(\text{“Assign tools to clusters”}, \{label_k\}_{k=1}^{K}, t)$
18:      $C_k \leftarrow C_k \cup \{t\}$
19:      SourceQuestionMapForCluster$[C_k].add($SourceQuestionMap$[t])$

                                                            ▷ **Phase 3: Tool Aggregation**
20: **for** $C_k \in \{C_1, \ldots, C_K\}$, where $C_k = \{t_{km}\}_{m=1}^{m_k}$ **do**
21:      Blueprint $\leftarrow$ Code Agent$(\text{“Design blueprint”}, C_k)$
22:      $C_k^* \leftarrow$ Code Agent$(\text{“Implement codes”}, C_k, \text{Blueprint})$
23:      **while** $turn <$ Max_Checking_Turns **do**
24:          all_feedback $\leftarrow$ Reviewing Agent$(C_k^*, $SourceQuestionMapForCluster$[C_k])$        ▷ Solve with $LLM_{\text{solver}}$
25:          **if** AllFeedbacksShowingPass(all_feedback) **then break**
26:          $C_k^* \leftarrow LLM(\text{“Refine codes”}, C_k, \text{all\_feedback})$
27:      $\mathcal{L} \leftarrow \mathcal{L} \cup \{C_k^*\}$
28: **return** $\mathcal{L}$

---

The reviewing agent is responsible for rigorously validating the functionality of the library. Its action space includes invoking $LLM_{\text{solver}}$, generating feedback, and making acceptance decisions. Specifically, it calls $LLM_{\text{solver}}$ to answer the source questions and collect corresponding reasoning trajectories. These trajectories are then analyzed to produce a structured report containing the decision (i.e., pass or requires refinement), the rationale behind this judgment, and detailed modification suggestions. This iterative cycle between the code agent and the reviewing agent continues until either the maximum number of iterations is reached or the reviewing agent determines that the library has covered all functionality in the initial tools. After applying this aggregation step to all clusters in $\mathcal{C}$, we obtain the structured Python library $\mathcal{L} = \{C_1^*, C_2^*, ..., C_K^*\}$, which can then be applied in tool-augmented reasoning. All prompts for our system are shown in Appendix E.

## 2.5 TOOL-AUGMENTED LLM REASONING

We adopt an embedding-retrieving strategy for tool-augmented LLM reasoning. First, we convert each tool $t^*$ in $\mathcal{L}$ into a standardized JSON format for function calling, which includes the function name, accepted parameters, and a description. We extract a textual representation $d_{t^*}$ for each tool by combining its name and description. We then encode each $d_{t^*}$ into a dense vector $\boldsymbol{v_{t^*}} = E(d_{t^*})$ using an embedding model $E$ to populate a vector index. When an LLM needs to use a tool during tool-augmented reasoning, it first generates a natural language searching query $sq$ describing the required functionality. This query is then encoded into a vector $\boldsymbol{v_{sq}}$, which retrieves the top-$k$ most relevant function candidates from the $\mathcal{L}$ through a k-Nearest Neighbor (k-NN) (Peterson, 2009) search. The retrieved subset of functions is then presented to the LLM with a standardized JSON format. Based on the given task and the retrieved candidates, the LLM actively selects the most appropriate function to use, analyzes the execution result from the tool, and then returns the final

answer. Specifically, we explicitly prompt the LLM not to overly trust the retrieved candidates, as the tools are not always appropriate or helpful for the given task.

## 3 EXPERIMENTS

### 3.1 EXPERIMENTAL SETUP

**Datasets and Models for Library Making**   Our experiments are conducted across multiple reasoning domains to evaluate the generalizability of TOOLLIBGEN. We constructed three libraries for different professional domains: a Science Library (from Llama-Nemotron-Post-Training-Dataset (Bercovich et al., 2025)), a Mathematics Library (from DeepMath-103K (He et al., 2025)), and a Medical QA Library (from ReasonMed (Sun et al., 2025)). More details about the dataset processing are in Appendix B. To generate the tool library, we employed GPT-5 (OpenAI, 2025b) as the general LLM and GPT-4.1 (OpenAI, 2025a) as $LLM_{solver}$ in our pipeline. We note that the same libraries are used to evaluate other LLMs as well.

**Evaluation Setting**   We have two different evaluation settings: (1) **Seen Case Performance:** This setting focuses on whether a well-structured library makes it easier to retrieve useful tools for problems used in tool making. Specifically, we use the same dataset $\mathcal{D}$ for both tool making and testing. (2) **Unseen Case Performance:** The second setup aims to measure the generalization ability when facing new problems. We use the SuperGPQA dataset (Du et al., 2025) as our test set, which contains questions from the scientific, mathematical, and medical domains. In our evaluation, we test the specific domain with the specific library, e.g., only use the math library for math question answering. Although this dataset shares similar concepts to the library-making datasets we use (which thus shows the potential for it to be better addressed if our tool library is properly formed), the format of the problems and the difficulty level differ from those of the library-making datasets. Therefore, we consider this dataset to be out of distribution. For both settings, the Accuracy of a model is calculated based on exact match for the multiple-choice questions and simple-evals [1] for numerical answers. More details of the evaluation case selection are in the Appendix B.

**Baselines**   We compare our approach with the following baselines: (1) **CoT**: Standard prompting to elicit reasoning steps in natural language. (2) **Program-of-thought (PoT)** We equip the LLM with a Python interpreter as the only tool (Chen et al., 2022; Gao et al., 2023). (3) **Fragmented Toolset (FT)**: After generating all question-specific tools, we do not cluster or aggregate them; instead, we directly retrieve from the fragmented toolset at inference time. (4) **Clustered Toolset (CT)**: After generating all tools, we cluster them but do not aggregate them. At inference time, the LLM first predicts the relevant cluster and then retrieves the appropriate tools within that cluster. (5) **KTCE**: A prior work to iteratively optimize the toolset with function operation, such as mutation and crossover (Ma et al., 2025). We perform experiments using a range of LLMs, including GPT-4.1 (OpenAI, 2025a), GPT-oss-20B (OpenAI, 2025c), and Qwen3-8B (Yang et al., 2025). For tool retrieval, we use SFR-Embedding (Meng et al., 2024) and retrieve the top-5 tools every time.

### 3.2 MAIN RESULT

Table 1 presents the results of our two setups of evaluations, which suggest that:

**Providing useful tools helps question-answering**   We find that augmenting LLM reasoning with tools improves question-answering performance only when the tools encode genuinely useful knowledge or methods. Compared with standard CoT, incorporating such specialized tools leads to better performance, even on unseen tasks. However, simply providing a code interpreter is not sufficient, as PoT does not always outperform CoT. This indicates that the key factor is not the availability of code execution capabilities, but the actual utility of the knowledge embedded in the tools for solving the underlying reasoning task.

**Our method outperforms baselines in seen cases.**   In seen cases, where the problems have been previously encountered, the tool library is more likely to contain a relevant and effective tool for each problem. The primary challenge for an LLM is to find and utilize it. Our method excels over all other baselines by 5%-10% improvement. This significant gain highlights that our approach helps

---

[1] https://github.com/openai/simple-evals

Table 1: Accuracy (%) of each method across different reasoning tasks.

| Method | Retriever | Seen Case | | | | Unseen Case in SuperGPQA | | | |
| --- | --- | --- | --- | --- | --- | --- | --- | --- | --- |
| | | Nemotron | DeepMath | ReasonMed | Average | Science | Math | Medicine | Average |
| Solver LLM: GPT-4.1 | | | | | | | | | |
| CoT | ✗ | 55.8 | 49.3 | 66.7 | 55.9 | 48.6 | 67.0 | 55.8 | 57.1 |
| PoT | ✗ | 52.4 | 48.5 | 59.8 | 52.6 | 44.6 | 63.2 | 48.8 | 52.2 |
| FT | ✓ | 64.7 | 52.8 | 71.4 | 64.0 | 48.8 | 68.2 | 54.6 | 57.2 |
| CT | ✓ | 68.8 | 54.3 | 73.4 | 65.5 | 49.4 | 69.3 | 57.4 | 58.7 |
| KTCE | ✓ | 67.9 | 57.6 | 72.9 | 66.1 | 49.1 | 69.5 | 56.9 | 58.5 |
| TOOLLIBGEN | ✓ | **75.9** | **59.3** | **75.6** | **70.3** | **51.3** | **71.4** | **59.1** | **60.6** |
| Solver LLM: GPT-oss-20B | | | | | | | | | |
| CoT | ✗ | 52.7 | 40.7 | 46.5 | 46.6 | 49.1 | 72.8 | 40.2 | 54.0 |
| PoT | ✗ | 47.5 | 41.4 | 45.6 | 44.8 | 47.3 | 69.1 | 38.5 | 51.6 |
| FT | ✓ | 60.6 | 48.7 | 55.6 | 55.0 | 51.3 | 73.1 | 47.6 | 57.3 |
| CT | ✓ | 62.6 | 49.6 | 55.1 | 55.8 | 51.7 | 74.1 | 45.6 | 57.1 |
| KTCE | ✓ | 68.4 | 48.8 | 57.3 | 58.2 | 52.5 | 74.0 | 44.8 | 57.1 |
| TOOLLIBGEN | ✓ | **74.3** | **52.7** | **59.4** | **62.1** | **53.5** | **76.1** | **51.8** | **60.5** |
| Solver LLM: Qwen3-8B | | | | | | | | | |
| CoT | ✗ | 41.5 | 38.1 | 52.8 | 45.5 | 46.5 | 69.6 | 42.4 | 52.8 |
| PoT | ✗ | 40.6 | 39.5 | 46.6 | 43.1 | 44.3 | 66.2 | 37.5 | 49.3 |
| FT | ✓ | 58.8 | 44.3 | 57.4 | 50.9 | 47.8 | 68.9 | 45.9 | 54.2 |
| CT | ✓ | 59.7 | 44.8 | 57.3 | 51.1 | 48.3 | 69.8 | 45.2 | 54.4 |
| KTCE | ✓ | 63.5 | 46.5 | 58.8 | 52.7 | 48.5 | 70.4 | 45.4 | 54.8 |
| TOOLLIBGEN | ✓ | **71.9** | **50.9** | **61.5** | **61.4** | **49.3** | **71.1** | **48.5** | **56.3** |

the effective retrieval process in finding useful tools, consistently making it easier for the LLM to access the right knowledge and tools to solve problems.

**Our method even demonstrates improved performance on unseen cases.** Compared with seen cases, unseen cases are much harder because the model has no prior experience with these specific problems, and there is no guarantee that a suitable tool exists in the library to solve them. Incorporating a fragmented toolset does not consistently enhance performance on unseen tasks (e.g., on the Math task with Qwen3-8B), as irrelevant tool information will hinder the problem-solving process. In contrast, our method consistently improves the performance 2-3% across different domains, which shows that our method offers suitable tools that supply effective information for this scenario.

### 3.3 ANALYSIS

**Why is our aggregated library better than the fragmented toolset?** In this section, we conducted a study to analyze why our aggregated library shows better performance. We attribute this to the retrieval issues with the fragmented toolset. In case of retrieving from the fragmented toolset, for each question $Q_i$, we consider the fragmented tools created from the question as the ground truths. Similarly, for retrieving from TOOLLIBGEN-produced tool library, we consider the aggregated counterparts of the extracted fragmented tools as the ground truths. A retrieval is considered successful when the LLM is able to retrieve at least one tool from the ground truths (although in practice, most questions correspond to exactly one tool). The questions are randomly selected from DeepMath-103k, and we use GPT-4.1 to generate the search query.

As shown on the left of Figure 3, the retrieval accuracy for a fragmented tool declines sharply as the number of tools increases. This demonstrates that an unorganized tool set fails to scale due to retrieval limitations. In contrast, our aggregated library maintains a high success retrieving accuracy by grouping functionally similar tools. A case study is shown on the right of Figure 3. For a question about an unfair die, the LLM generates a search query to retrieve tools. The retrieved tool from the fragmented toolset is related, but does not contribute meaningfully to solving the unfair die problem. By contrast, the retrieved tool from our library is a general tool for binomial questions and provides all the information required for the unfair die scenario. Therefore, the advantages of our library stem from merging superficially related yet functionally overlapping tools, thereby minimizing the likelihood of retrieving relevant but unhelpful tools.

**Ablation Study** We conducted a thorough ablation study to validate the design of each module in TOOLLIBGEN. For both question-specific tool creation and aggregation, we confirmed the neces-

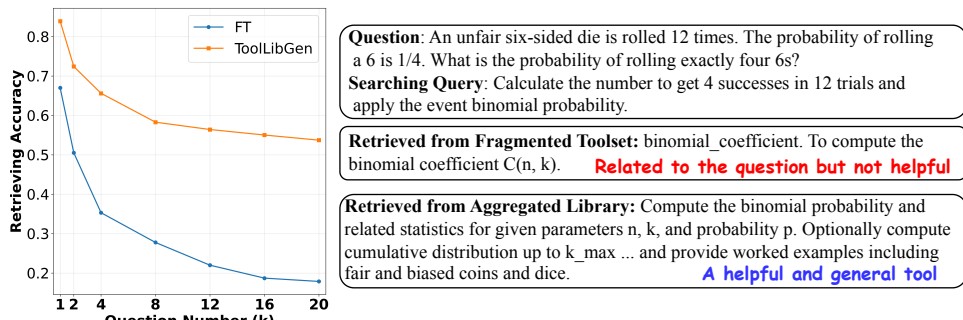

Figure 3: (Left) The retriever accuracy changes as the number of questions for tool-making increases from 1k to 20k. (Right) A case study showing how TOOLLIBGEN facilitates tool retrieval.

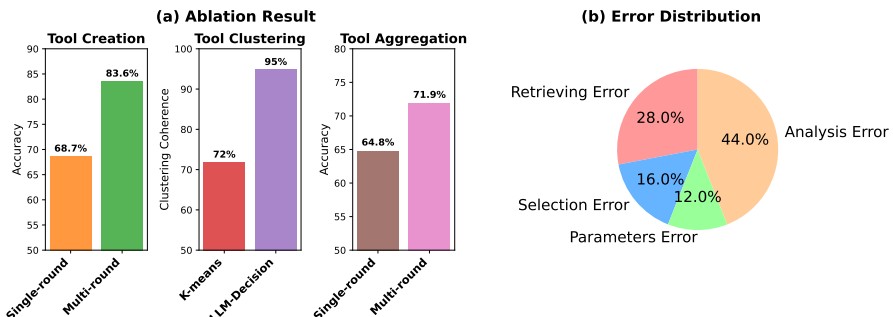

Figure 4: (a) Ablation results showing the effectiveness of our design; (b) Error distribution of GPT-4.1 in reasoning augmented by tools generated by TOOLLIBGEN.

sity of our iterative validation by comparing our performance against a single-pass alternative. For tool clustering, we demonstrated the superiority of our LLM-based method over a standard K-means baseline through manual evaluation of cluster coherence. More setup details are in the Appendix C. As shown in Figure 4, these results highlight the crucial role of multi-turn revision and the effectiveness of leveraging an LLM for high-quality tool clustering.

**When does the library fail to help the reasoning?** We conducted an error analysis of GPT-4.1's errors in tool-augmented reasoning based on TOOLLIBGEN. From DeepMath-103k, we sampled 50 erroneous tool-augmented cases for analysis. The error types can be divided into: (1) *Retrieving Error*: the error was due to failing to retrieve useful tools; (2) *Selection Error*: retrieval succeeded but the LLM failed to select the appropriate tool among the retrieved ones; (3) *Parameters Error*: the LLM supplied incorrect parameters when using the tool; (4) *Analysis Error*: the LLM over-relied on its own beliefs during reasoning and did not leverage the tool's information. The error types are shown in the Figure 4. The most significant challenge is the *Analysis Error*. This arises because many problems are complex, and our tools may not be able to provide a direct answer. They can offer effective analyses to help the LLM in reasoning. However, the LLM often fails to leverage these analyses and instead produces an incorrect result based on its own belief. This demonstrates that although tools can assist the LLM by providing analysis, there is still a critical need to enhance the LLM's inherent capabilities to handle complex problems.

**Can we teach an LLM to propose a better search query via supervised-finetuning?** We explore whether we could improve an LLM's tool-augmented reasoning by teaching it to ask a better search query during retrieval. We designed a data collection method using 5k randomly sampled examples from the Nemotron dataset's science domain (excluding the testing data used in Table 1) to obtain higher-quality search queries. For each tool $t$ derived from a question $Q$, we trace the aggregated tool $t^*$ and pair the question $Q$ with the aggregated tool $t^*$. Subsequently, we prompt Qwen3-8B to roll out different searching queries based on question $Q$ and only keep the query that can correctly recall $t^*$. Subsequently, we fine-tune Qwen3-8B with the correctly retrieved trajectory. After fine-tuning, the accuracy of Qwen3-8B on the Nemotron dataset reached 72.8%, a slight

improvement over the 71.9% accuracy without this training (as shown in Table 1). We believe this marginal gain is because the LLM struggles to learn the nuances of effective query formulation through supervised-finetuning alone, as most queries it generates appear reasonable even without training. Future work could benefit from introducing hard negative examples to help the model better distinguish the nuanced difference between queries.

## 4 RELATED WORK

**Tool-Augmented LLM Reasoning**   LLMs have evolved significantly from relying solely on natural language reasoning to tool-augmented reasoning. For tool-augmented reasoning, LLMs are empowered to invoke external modules, such as code interpreters, search engines, and specialized APIs (Mialon et al., 2023; Xu et al., 2023; Qu et al., 2025). For example, SciAgent (Ma et al., 2024) has demonstrated that by equipping LLMs with domain-specific toolsets, their method can improve an LLM's performance on complex scientific reasoning tasks. Some studies have found that directly having an LLM use code can improve its reasoning performance and calibration ability (Chen et al., 2022; Gao et al., 2023; Yue et al., 2023), while other works have shown that LLMs can learn to use Python code at the appropriate time to solve mathematical problems (Yue et al., 2024; Li et al., 2025). However, the widespread adoption of tool-augmented reasoning is constrained not just by the availability of domain-specific tools, but by the absence of a scalable methodology for their management.

**Tool Creation for LLM Reasoning**   To address the scarcity of domain-specific tools, recent research has explored methods for their automatic creation (Zhang et al., 2024; Zheng et al., 2025; Wang et al., 2025). Seminal works like CREATOR (Qian et al., 2023) and CRAFT (Yuan et al., 2023) established the feasibility of this paradigm by prompting LLMs to abstract reusable Python functions from problem-solving traces. Subsequent research has acknowledged this organizational problem and proposed various curation strategies. Some methods focus on labeling semantically similar tools or skills without refactoring the underlying code (Didolkar et al., 2024). Otherwise, TroVE (Wang et al., 2024) induces a toolbox through a continuous cycle of growth and trimming, ReGAL (Stengel-Eskin et al., 2024) applies iterative code refactoring to each tool to conduct the abstractions, and KTCE (Ma et al., 2025) performs localized, pairwise merging of tools. While these methods represent significant progress, they often process tools individually and lack a global design principle for all tools. Such a global design moves beyond simple function merging to design and implement unified, object-oriented structures such as Python classes, which encapsulate the shared functionality of an entire tool group. By transforming a fragmented set of functions into a structured, well-organized library, it can address the critical bottlenecks of retrieval efficiency and semantic ambiguity, enabling automated tool creation to scale in a robust and sustainable manner.

## 5 CONCLUSION

In this work, we introduced TOOLLIBGEN, a pipeline for transforming a fragmented toolset into a structured Python library through clustering and a multi-agent refactoring aggregation. Our experiments across multiple domains and foundation models demonstrate that the aggregated library design significantly improves task accuracy compared with the fragmented toolset. This highlights the importance of shifting the focus from isolated tool generation to systematic organization and abstraction, which better supports scalable and reliable tool-augmented reasoning. Our study highlights how data can be more effectively leveraged to enhance downstream reasoning performance without additional training. Future work can explore the co-evolution of tool creation, tool aggregation, and tool learning, enabling LLMs to simultaneously create reusable tools and refine their usage strategies, thereby continually extending their reasoning capabilities.

## 6 REPRODUCIBILITY STATEMENT

We have carefully documented all implementation details and experimental setups in the main text and the appendix sections to ensure transparency. In particular, all prompts used throughout our pipeline are provided in Appendix E, and additional dataset processing details are included in Appendix B. The complete implementation code will be released, along with the constructed tool libraries used in our experiments, so that future researchers can fully reproduce and extend our work.

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

## A  THE USE OF LARGE LANGUAGE MODELS

In this project, we leveraged a LLM in paper polishing. We understand and accept full responsibility for all content, including any text generated with the assistance of an LLM. We have reviewed all such contributions to ensure their accuracy and originality.

## B  EXPERIMENT DETAILS

### B.1  DATASET

For the Nemotron dataset, we first performed data cleaning, which included removing duplicate questions and using GPT-5 to evaluate question quality in order to filter out under-specified problems. In terms of topic filtering, we only retained questions related to physics, chemistry, and biology. Since Nemotron does not guarantee the correctness of the original answers, we re-answered the questions using GPT-5 and kept only those whose answers were consistent with the original ones. For DeepMath-103k, we used all 103k questions. For ReasonMed, we randomly sampled 100k questions. Since the correctness of the original answers could not be guaranteed, we applied the same strategy as with Nemotron: GPT-5 was used to re-answer the questions, and only those consistent with the original answers were retained. The number of questions, tools of fragmented toolset and tools of our library are shown in Table 2.

|  | # of Question | # of Question-Specific Tool | # of Library Tool |
|---|---|---|---|
| Science | 33k | 48k | 3.1k |
| Math | 103k | 175k | 8.9k |
| Medical | 100k | 142k | 5.8k |

Table 2: The number of toolset of different domains

During the testing phase, for the seen cases, most questions from the Nemotron and ReasonMed training sets were relatively straightforward, allowing the LLM to answer them directly without tools. This made it difficult to effectively compare the performance of different tool aggregation strategies, as variations in strategy had little impact on the final outcome. To address this, we first performed CoT inference on all cases. From the cases where the CoT inference failed, we randomly sampled 1k questions as hard questions from each dataset. It is important to note that these are multiple-choice questions; when solving these hard examples, the LLM might have made a wrong choice between two plausible options, leading to the first time CoT inference failure. In subsequent sampling, the LLM might happen to guess the correct option, which is why the CoT success rate isn't zero. Given the inherently high difficulty of the DeepMath dataset, we directly and randomly sampled 1k questions from it for our evaluation. For superGPQA, we use the questions with labels from physics, chemistry, and biology as science QA, math-labeled questions were used as math QA, and medicine-labeled questions were used as medical QA.

### B.2  HYPER-PARAMETERS

When we perform clustering, we initially set the number of seed tools to 1k. We update in batches of 200, with a maximum of 500 update rounds. For the creation and aggregation phases, the maximum number of modification rounds is set to 3.

## C  ABLATION STUDY SETTING

To understand the contribution of each component of our framework, we conduct a thorough ablation study for each key component of our pipeline.

**Question-specific Tool Creation**  We first evaluated the necessity of our multi-round generation process, which incorporates validation from an $LLM_{solver}$. We compared this approach against a single-pass generation on a randomly sampled 300 questions from the subset of Nemotron used in testing, using Qwen3-8B as the $LLM_{solver}$. Performance was measured by the accuracy of the

generated tool in solving the corresponding question. The results show that the multi-round process yields higher accuracy, confirming that iterative validation and refinement are crucial for creating effective tools.

**Tool Clustering**  Next, we compared the quality of our LLM-based clustering method against a standard K-means baseline. For the baseline, we applied K-means to SFR-embeddings derived from each tool's title and description, setting the number of clusters (K) to be identical to the number of leaf nodes generated by our method. To evaluate, we randomly sampled 100 tools and retrieved three other tools from the same cluster using both techniques. A clustering case was deemed correct only if manual inspection confirmed that all three retrieved tools belonged to the same functional sub-domain. Our method achieved 95% accuracy, significantly outperforming the K-means baseline, which only reached 72%, demonstrating its superior ability to create coherent tool groups.

**Tool Aggregation**  Finally, for tool aggregation, we similarly validated the effectiveness of the multi-round generation and validation process. Following the same experimental setup as in the tool creation study, we observed that a single-pass approach is often insufficient, as the LLM fails to abstract all reasoning scenarios covered by the tools, leading to critical omissions. The multi-turn revision process proved essential for significantly improving the aggregated library's final performance.

## D  EXAMPLE

**Question 1:**
A particle moves along a straight line with its velocity as a function of time given by $v(t) = 5t^2 + 2t$. What is its acceleration function, a(t)?

**Tool 1:**

```
def apply_bayes(likelihood_A: float, prior_A: float,
likelihood_not_A: float, prior_not_A: float) -> float:
    # Calculate the numerator of Bayes' theorem: P(B|A) * P(A)
    numerator = likelihood_A * prior_A

    # Calculate the total probability of the evidence B (the denominator):
    # P(B) = P(B|A)P(A) + P(B|not A)P(not A)
    marginal_likelihood = (likelihood_A * prior_A)
    \+ (likelihood_not_A * prior_not_A)

    # Avoid division by zero if the evidence is impossible
    if marginal_likelihood == 0:
        return 0.0

    # Compute the posterior probability
    posterior_A = numerator / marginal_likelihood

    return posterior_A
```

**Question 2:**
A mother with blood type O and a father with blood type AB have twins, both sons, with blood type B. Given that 32% of all twins have different genders, calculate the probability that the twins are identical.

**Tool 2:**

```
from collections import Counter
import json

def get_abo_offspring_allele_distribution
(mother_genotype: str, father_genotype: str) -> dict:
```

```
6      # Get alleles from each parent
7      mother_alleles = list(mother_genotype)
8      father_alleles = list(father_genotype)
9
10     # Generate all possible offspring allele combinations
11     possible_genotypes = []
12     for m_allele in mother_alleles:
13         for f_allele in father_alleles:
14             # Sort to standardize (e.g., AO not OA)
15             genotype = ''.join(sorted((m_allele, f_allele)))
16             possible_genotypes.append(genotype)
17
18     # Count occurrences of each genotype
19     genotype_counts = Counter(possible_genotypes)
20
21     # Calculate probabilities
22     total_combinations = len(possible_genotypes)
23     genotype_distribution = {geno: count / total_combinations
24     for geno, count in genotype_counts.items()}
25
26     return genotype_distribution
```

**Question 3:**

You have two coins: a fair coin with a probability of heads as 0.5, and a biased coin with a probability of heads as 0.8. You randomly select one of these coins and flip it repeatedly, obtaining heads each time. Determine the minimum number of consecutive heads required such that the probability of the coin being fair is less than 0.1.

**Tool 3:**

```
1  def equal_priors(items):
2      """
3      Assign equal prior probability to each item in the list.
4      Returns a dict mapping item -> 1/len(items).
5      """
6      n = len(items)
7      if n == 0:
8          return {}
9      p = 1.0 / n
10     return {item: p for item in items}
11
12 def apply_bayes(likelihood_A: float, prior_A: float,
13             likelihood_not_A: float, prior_not_A: float) -> float:
14     """
15     Returns P(A|B) using Bayes' theorem for a binary hypothesis.
16     """
17     num = likelihood_A * prior_A
18     denom = num + likelihood_not_A * prior_not_A
19     if denom == 0:
20         raise ZeroDivisionError("Total probability is zero; check inputs.")
21     return num / denom
```

**Aggregated Tool:**

```
1  from typing import List, Dict, Any, Union
2  import math
3  from abc import ABC, abstractmethod
4
5  class Hypothesis:
6      """Represents a single hypothesis with a name and a prior probability.
        """
7      def __init__(self, name: str, prior: float):
8          if not (0.0 <= prior <= 1.0):
```

```python
            raise ValueError("Prior probability must be between 0 and 1.")
        self.name = name
        self.prior = prior

    def __repr__(self) -> str:
        return f"Hypothesis(name='{self.name}', prior={self.prior:.4f})"

class HypothesisSet:
    """Manages a set of mutually exclusive hypotheses, ensuring priors sum
        to 1."""
    def __init__(self, hypotheses: List[Hypothesis]):
        total_prior = sum(h.prior for h in hypotheses)
        if not math.isclose(total_prior, 1.0, rel_tol=1e-9):
            raise ValueError(f"Priors must sum to 1, but they sum to {
                total_prior}")
        self._hypotheses = {h.name: h for h in hypotheses}

    def get(self, name: str) -> Hypothesis:
        return self._hypotheses[name]

    @property
    def names(self) -> List[str]:
        return list(self._hypotheses.keys())

    def __iter__(self):
        return iter(self._hypotheses.values())

class BaseEvidence(ABC):
    """Abstract base class for all forms of evidence."""
    @abstractmethod
    def get_likelihoods(self, hypothesis_set: HypothesisSet) -> Dict[str,
        float]:
        """
        Returns a dictionary mapping each hypothesis name to its likelihood
        P(Evidence | Hypothesis).
        """
        pass

class DirectLikelihoodEvidence(BaseEvidence):
    """
    Evidence defined by direct likelihood values (can be probabilities or
        densities).
    This class solves the problem of densities > 1.
    """
    def __init__(self, likelihood_map: Dict[str, float]):
        for name, val in likelihood_map.items():
            if val < 0:
                raise ValueError(f"Likelihood for '{name}' cannot be negative
                    .")
        self._likelihood_map = likelihood_map

    def get_likelihoods(self, hypothesis_set: HypothesisSet) -> Dict[str,
        float]:
        if not set(self._likelihood_map.keys()) == set(hypothesis_set.names
            ):
            raise ValueError("Likelihood map must contain keys for all
                hypotheses.")
        return self._likelihood_map

class RepeatedSuccessEvidence(BaseEvidence):
    """
    Evidence from 'n' consecutive successes, where the success probability
    depends on the hypothesis.
    """
```

```python
    def __init__(self, n_successes: int, success_prob_map: Dict[str, float
        ]):
        if n_successes < 0:
            raise ValueError("Number of successes cannot be negative.")
        self.n_successes = n_successes
        self.success_prob_map = success_prob_map

    def get_likelihoods(self, hypothesis_set: HypothesisSet) -> Dict[str,
         float]:
        likelihoods = {}
        for name in hypothesis_set.names:
            p = self.success_prob_map[name]
            likelihoods[name] = p ** self.n_successes
        return likelihoods

class BayesianSolver:
    """The engine that calculates posterior probabilities from hypotheses
         and evidence."""
    def __init__(self, hypothesis_set: HypothesisSet):
        self.hypothesis_set = hypothesis_set
        self.evidence_list: List[BaseEvidence] = []

    def add_evidence(self, evidence: BaseEvidence):
        self.evidence_list.append(evidence)

    def calculate_posterior(self) -> Dict[str, float]:
        """
        Calculates the posterior probabilities using log-space for
            numerical stability.
        """
        # Start with log of priors
        log_posteriors = {h.name: math.log(h.prior) for h in self.
            hypothesis_set}

        # Accumulate log likelihoods from all evidence
        for evidence in self.evidence_list:
            likelihoods = evidence.get_likelihoods(self.hypothesis_set)
            for name in self.hypothesis_set.names:
                # Use a small epsilon to avoid log(0)
                log_posteriors[name] += math.log(likelihoods[name] + 1e-300)

        # Normalize to get posterior probabilities
        # (This is a stable way to convert log probabilities back to a
            distribution)
        max_log_p = max(log_posteriors.values())
        posteriors_unnormalized = {name: math.exp(log_p - max_log_p) for
            name, log_p in log_posteriors.items()}
        total_p = sum(posteriors_unnormalized.values())

        if total_p == 0: # Handle case where all probabilities are zero
            num_hypotheses = len(self.hypothesis_set.names)
            return {name: 1.0 / num_hypotheses for name in self.
                hypothesis_set.names}

        final_posteriors = {name: p / total_p for name, p in
            posteriors_unnormalized.items()}
        return final_posteriors

def bayesian_scenario_solver(scenario_spec_json: str) -> Dict[str, Any]:
    """
    A high-level facade to model and solve various Bayesian scenarios from
         a JSON string.

    Args:
```

```
120        scenario_spec_json (str): A JSON string describing the entire
             problem.
121          - 'hypotheses': List of dicts, e.g., [{'name': 'H1', 'prior':
                 0.5}, ...]
122          - 'evidence': (Optional) List of evidence dicts. Each needs a '
                 type'.
123          - 'goal': (Optional) A dict describing a complex goal, like
                 finding a minimum 'n'.
124
125    Returns:
126        A dictionary containing the results of the analysis.
127    """
128    # 1. Parse the JSON string input into a Python dictionary
129    try:
130        scenario_spec = json.loads(scenario_spec_json)
131    except json.JSONDecodeError as e:
132        raise ValueError(f"Invalid JSON format: {e}")
133    hyp_defs = [Hypothesis(h['name'], h['prior']) for h in scenario_spec['
           hypotheses']]
134    hypothesis_set = HypothesisSet(hyp_defs)
135
136    # 2. Check for a complex goal (like for Problem 3)
137    if 'goal' in scenario_spec and scenario_spec['goal']['type'] == '
           find_minimum_n':
138        goal = scenario_spec['goal']
139        n = 1
140        while True:
141            solver = BayesianSolver(hypothesis_set)
142            evidence = RepeatedSuccessEvidence(
143                n_successes=n,
144                success_prob_map=goal['success_prob_map']
145            )
146            solver.add_evidence(evidence)
147            posteriors = solver.calculate_posterior()
148
149            target_posterior = posteriors[goal['hypothesis_to_track']]
150
151            condition_met = False
152            if goal['condition'] == '<' and target_posterior < goal['
                 threshold']:
153                condition_met = True
154            elif goal['condition'] == '>' and target_posterior > goal['
                 threshold']:
155                condition_met = True
156
157            if condition_met:
158                return {
159                    'status': 'Goal Reached',
160                    'result': {'minimum_n': n},
161                    'final_posteriors': posteriors
162                }
163            n += 1
164            if n > 1000: # Safety break
165                raise RuntimeError("Exceeded 1000 iterations without reaching
                     goal.")
166
167    # 3. Standard posterior calculation (for Problems 1 & 2)
168    solver = BayesianSolver(hypothesis_set)
169    if 'evidence' in scenario_spec:
170        for ev_spec in scenario_spec['evidence']:
171            if ev_spec['type'] == 'direct_likelihoods':
172                evidence = DirectLikelihoodEvidence(likelihood_map=ev_spec['
                     values'])
173                solver.add_evidence(evidence)
174            # Other evidence types could be added here
```

```
175            else:
176                raise ValueError(f"Unknown evidence type: {ev_spec['type']}")
177
178    posteriors = solver.calculate_posterior()
179    return {
180        'status': 'Calculation Complete',
181        'result': {'posteriors': posteriors}
182    }
```

# E PROMPT

---

**Tool Generation Prompt**

You are an expert in creating tools. Below is the single worked example we have:
**Question:**
{question}
**CoT Answer:**
{CoT Process}
**YOUR TASK:**
Write one or multiple general tool that can solve similar sub question. Each tool must be an executable Python function designed to perform one atomic reasoning step.
**Output Format:**
For the question, generate one general tools, use:
- Each tool must be a executable Python function designed to the specific question.
- Make the Python function general: any database or tool should work with minimal tweaks. Only provide code for steps that truly benefit from automation.
- No code is needed for selecting the final answer; toolkits are for supporting intermediate reasoning only.
- Try to save knowledge into static variables; don't just think of the calculation process as a tool, but also think of the knowledge sentences as tools;
- For example, for the question: Can an object's momentum change without experiencing net acceleration? No calculation codes are needed, but knowledge sentences are needed for analysis. Even just some hints in the Python function is fine, but you should make sure that the knowledge is saved into static variables.
- In this phase, the format is:

```
<tool1>
def function_name(param_1_name, param_2_name,...):
[Implement a general function for one sub-question]
return f"explain the result: {{result}}"
<\code><\tool1><tool2>...<\tool2>
```

---

**Tool Verification Prompt**

You are an expert problem solver who uses available tools to analyze and solve questions.
**Question to solve:**
{question}
**Available Tools:**
{tools description}
**Your Task:**
1. Use the available tools systematically to help analyze and solve the question;
2. Make sure to actually use the tools in your analysis process;
3. Provide clear reasoning for each step;
4. Give a final answer based on your tool-assisted analysis;
**Final Response Format:**

```
<analysis>
[Show tool outputs and explain how they contribute
to your analysis]</analysis>
<answer>Your conclusive answer</answer>
```

---

**Tool Refinement Prompt**

You are an expert tool effectiveness evaluator.
**Original Question:** {question}
**Generated Tools:** {tools}
**Ground Truth Answer:** {answer}
**Tool Usage Analysis Results:** {evaluation_results}
**Your Task:** Analyze whether the generated tools were truly effective for solving the question:
1. Tool Utility Assessment:
- Did the tools provide meaningful assistance in solving the question?
- Were the tools actually used in the solution process?
- Did the tools contribute to reaching the correct answer?
2. Tool Quality Assessment:
- Are the tools properly implemented and functional?
- Do the tools address the core reasoning steps needed for this type of question?
- Are the tools generalizable to similar questions?
3. Code Refinement:
Based on the effectiveness analysis, refine the tools to make them more useful for solving the question.
**Response Format**
[Detailed explanation of why tools are effective or ineffective]
[If ineffective, specific suggestions for improvement]

```python
<tool1>
```python
def function_name(param_1_name, param_2_name,...):
[Implement a general function for one sub-question]
return f"explain the result: {{result}}"
</tool1><tool2>...</tool2>...
```

---

**Cluster Proposing Prompt**

You need to aggregate the following tools into a hierarchy. You do not need to set each tool in different nodes. In contrast, you should set the tools that are similar to each other in the same node. Please make sure that the hierarchy should not be too shallow. Deep hierarchy and detailed classification are preferred. The depth of the hierarchy should be {cluster_depth}.
The function_name of tools should NOT be the last layer leaf node of the hierarchy. Do not need to include the function_name in the hierarchy. tools:
{tool_lst}
Your output should be a JSON object with a hierarchical structure.

```
{{
  "clusters": [ {{id, level, parent, children}} ... ],
  }}
example:
{{
  "clusters": [
    {{
      "id": "c_root",
      "level": 0,
      "parent": null,
      "children": ["c_math", "c_utils",...]
    }},
    {{
      "id": "c_math",
      "level": 1,
      "parent": "c_root",
      "children": ["c_arith", "c_stat",...]
    }},
    {{
      "id": "c_linear_algebra",
      "level": 2,
      "parent": "c_math",
      "children": ["c_matrix","c_singular_value_decomposition",...],
    }},
```

```
    {{
      "id": "c_stat",
      "level": 2,
      "parent": "c_math",
      "children": ["c_stat_mean", "c_stat_t_test",...],
    }},
  ]
}}
```
DELIVERABLE Return ONLY the JSON array described in OUTPUT FORMAT. No any other text. No '''json.

---

## Cluster Update Prompt

Given the current hierarchy and new tools to integrate, generate specific operations to update the hierarchy incrementally instead of rewriting the entire structure.
Analyze the new tools and determine what changes are needed:
1. ADD_NODE: Create new clusters for tools that don't fit existing categories
2. MODIFY_NODE: Update existing cluster properties if needed
3. No operations if tools fit perfectly into existing leaf nodes
Current hierarchy:
current_hierarchy
New tools to integrate:
tool_lst
Your output should be a JSON object with specific operations:

```
{{
  "operations": [
    {{
      "action": "ADD\_NODE",
      "node\_id": "new\_cluster\_id",
      "level": ...,
      "parent": "parent\_cluster\_id",
      "description": "Description of what this cluster represents",
      "reasoning": "Why this new cluster is needed for the new tools"
    }},
    {{
      "action": "MODIFY\_NODE",
      "node\_id": "existing\_cluster\_id",
      "changes": {{
        "add\_children": ["new\_child\_id1", "new\_child\_id2"]
      }},
      "reasoning": "Why this modification is needed"
    }}
  ]
}}
```
Guidelines:
- Only create new nodes when new tools represent significantly different functionality
- Prefer adding to existing leaf nodes when tools are similar enough
- Maintain proper parent-child relationships and level consistency
- Keep the hierarchy depth appropriate (not too shallow, not too deep)
- Provide clear reasoning for each operation
DELIVERABLE Return ONLY the JSON array described in OUTPUT FORMAT. No any other text. No '''json.

---

## Blueprint Design Prompt

Persona:
You are a Senior Python Library Architect; you transform fragmented helper functions into coherent, maintainable knowledge-libraries by applying rigorous knowledge-engineering practice.
Your Mission:

You will receive a list of Python functions as discrete tools that all belong to the same sub-domain. Design a **Refactoring Blueprint** that reorganises those tools into a catalogue of **Static Inference Blocks (SIBs).**

Static Inference Block (SIB) Definition

A SIB is a reusable knowledge capsule that is composed of one or multiple Python classes to construct one specifc problem-solving scenario and multiple public function to wrap up all the functionality of the collected discrete tools. The def function can accept multiple parameters and return a multi-paragraph explanation string that follows the standard template shown below.

It should follow the following requirements: • accepts a well-defined set of input facts (pre-conditions)
• instantly infers deterministic outputs (formulae, numbers, long–form explanations)
• groups as many original tools as logically coherent—functionality must remain complete and loss-less.
Example of SIB:

```
# Given tool code 1:
def calculate_potential_energy(mass, height):
    Gravity = 9.81
    return mass * Gravity * height

# Given tool code 2:
def get_gravity(planet_name):
    if planet_name == "Earth":
        return 9.81
    elif planet_name == "Mars":
        return 3.71
    else:
        return 0

# The SIB should be composed of the following one class
and one public function:
class _PlanetaryPhysics:
    def __init__(self, planet_name):
        self.planet_name = planet_name
        _GRAVITY_MAP = {{
            "Earth": 9.81,
            "Mars": 3.71
        }}
        self.gravity = _GRAVITY_MAP[planet_name]
    def calculate_potential_energy(self, mass, height):
        return mass * self.gravity * height
    def get_gravity(self):
        return self.gravity

def calculate_potential_energy(mass, height, planet_name="Earth"):
    planetary_physics = _PlanetaryPhysics(planet_name)
    return planetary_physics.calculate_potential_energy(mass, height)
```

**Core Requirements** 1. High Cohesion / Low Coupling: Group functions that operate on the same core data or represent the same conceptual scenario. SIBs should be self-contained and independent.
2. Scenario-Focused Classes: Each SIB Class should just model one specific, concrete scenario (e.g., _ProjectileMotion, _GasContainerState). The class is initialized with the base facts of the scenario, from which all other properties can be derived. But the scenario should be specific, not too general.
3. Lossless Abstraction: The public execute function must provide a way to access the full functionality of all original tools it covers. Use optional parameters to control which specific calculations are performed. But the parameters should be simple to understand and use.
4. Descriptive Naming: SIB titles and class names must be explicit and descriptive. Avoid vague, generic umbrella names such as PhysicsToolkit or MathHelpers.
5. Blueprint, Not Code: Your output is a design document (the blueprint), not the final Python code.
6. All parameters should be simple to understand and use. No any hard-encoded condition in the parameters, e.g., "if parameter == a, then use function1, else use function2,". In this case, you need to create more than one public function to cover all the scenarios rather than using hard-encoded condition.
Output Format: The Refactoring Blueprint
Your entire output must be a single markdown document. This document will contain multiple SIB descriptions (depending on how many scenario you can infer from the current tools), each strictly adhering to the following metadata template.

Mandatory SIB Metadata (document for *every* SIB): [SIB]Title [Description] (Provide a concise, high-level summary in plain language describing what this SIB is and what problem it solves.) [SIB Class Description] (Design one or more classes. Each class should model one scenario where, given a few initial inputs, many other properties can be deterministically calculated. Describe the purpose of each class and what initial state it will be constructed with. Output every class's init function, inner function name, description and parameters.) [Public Function Description] (Describe one or more public functions. Detail which parameters it should accept to ensure all scenarios and functionalities of the covered tools are accessible. [Covered Tools] (List all the tool indices that are covered by this SIB, separated by commas. E.g., 1, 2, 3)

**Input: All Python Function Tool Code:** {tool_code_list}

**Deliverable:** Now start to write the blueprint. Your final output is the complete Refactoring Blueprint in markdown format. Do not include any Python implementation code. Exactly follow the format:

```
<SIB>
[SIB1_name]:...
[Description]...
[SIB Class Description]...
...
</SIB>
<SIB>
[SIB2_name]: ...
</SIB>
...
```

---

### Aggregation Code Implementation Prompt

Given the blueprint as input, write Python code that strictly adheres to the design and specifications presented in the blueprint. You must implement the class and the public execute function exactly as described, ensuring that all functionality, methods, and logic are fully covered.

1. Do not skip or simplify any part of the implementation.

2. All static variables must be included in the code with the same names and initial values (if specified).

3. The name of the public functions must be exactly showing the functionality of the tool. Each public function must have a Python Function Signature and a Google-style Docstring Args Section.

4. The public functions cannot use any nested params objects, no any kwargs like "params: Dict[str, Any]"; all parameters must be flattened into arguments, no any nested params objects.

5. The code could be very long, but you cannot refuse to generate the code because the code is very very important for improving global science knowledge.

6. Every public function should be independent, no any dependency between public functions. Because we will store each public function as a separate OpenAI tool, so the dependency between public functions will cause the tool to fail to call.

**A very special rules for the public functions:**

7. Input parameters MUST be limited to the following native types only: string, boolean, integer, array. If a parameter is a complex structure (object/dict, tuple, set, union, nested generics, or unknown composite), you MUST accept it as a string containing a valid JSON value. Never pass complex structures directly as Python objects.

8. JSON strings MUST be valid JSON: use double quotes, no comments, no trailing commas, properly escaped within the outer JSON; represent tuples as fixed-length arrays; represent sets as arrays (uniqueness handled in code).

9. For union types, choose one allowed JSON shape and encode only that shape as the JSON string (do not mix shapes). If unsure, default to the minimal valid example of the primary shape.

10. For optional parameters, omit the argument entirely if unused (do not send null or empty strings).

11. Each public function MUST include a Python function signature and a Google-style docstring Args section. Valid formats:
- Signature: `def func_name(param: Type = default, ...) ->
ReturnType:` on a single logical line (line breaks inside parentheses are allowed, but parameter tokens must follow `name: type` or `name=default` forms).
- Args entries (one per line), any of the following forms are accepted and will be parsed:
* `name (Type): description`
* `name: description`
* `- name (Type): description`
- Supported type hints for parsing include: `str—string`, `int—integer`, `float—number—double`, `bool—boolean`, `List[int—string]`.
- Any Dict[k,v], List[List[int]],... should be a string containing a valid JSON.

- Unsupported complex types such as 'Union[...]' should be documented in the description and passed as JSON string via a 'string' parameter.
Examples for complex parameters (to include in Args descriptions where applicable):
- 'data (string): Must be a valid JSON. Expected shape: {{"<category>": [[<int id>, "<name>"], ...]}}. Example: {{"fruits": [[1, "apple"], [2, "banana"]]}}'
- 'items (string): Must be valid JSON, either a JSON array of integers (e.g., [1,2,3]) or a JSON object of string→integer (e.g., {{"a":1,"b":2}}).'
For the comment in Google-style Docstring, you must:
1. Describe the function ability with a detailed description. At the end, it would be better to have some examples for this function to use, e.g., "This function can be used to calculate GCD of two numbers or LCM of two numbers." All things should be in one paragraph without any '

n'.
2. For each parameter, describe the parameter with a detailed description. Add at least one example for each parameter to show the parameter type and the expected value.
–Blueprint start–

{blueprint}

–Blueprint end–
Now start to generate the code according to the instructions: The output should start with  and end with . For all classes, start with <class> and end with </class>. For each public function, start with <function_{{index}}> and end with </function_{{index}}> (index from 1 to the number of public functions). This XML signal is important for the parser to parse the code correctly.

---

### Reviewing Agent Feedback Generation Prompt

You are analyzing whether the current tool library is helpful for a weaker LLM to solve problems.
**Original Question:** {question}
**Ground Truth Answer:** {ground_truth}
**Available Tools:** {tool_code}
**Weaker LLM's Complete Conversation:** {weaker_LLM_message}
**Weaker LLM's Final Answer:** {weaker_final_response}
**Your Task:** Analyze whether the weaker LLM was able to effectively use the provided tools to solve the problem. Consider: 1. Did the weaker LLM use the tools appropriately? 2. Did the tools provide sufficient functionality for the problem? 3. Was the final answer correct or close to the ground truth? 4. If errors occurred, what caused them and how can the tools be improved to prevent them? 5. What specific improvements would help the weaker LLM succeed?
**Important:** Even if errors occurred, focus on how to improve the tools to make them more robust and user-friendly for the weaker LLM.
You should only output the final report in the <final_report> tag.
<final_report>

```
\{\{
"is\_library\_helpful": "PASS" or "NEED\_PATCHING",
"reason": "Detailed analysis of the weaker LLM's performance and
tool usage. Include error analysis if applicable. Explain what
worked well and what didn't.",
"modification\_suggestions": "Specific suggestions for improving
the tools when NEED\_PATCHING. If errors occurred, explain how
to make tools more robust. Prioritize modifying existing functions
over adding new ones. Be very detailed and precise about what
changes would help the weaker LLM succeed."
\}\}
```

</final_report>

