# OpenReview forum: "ToolLibGen: Scalable Automatic Tool Creation and Aggregation for LLM Reasoning"
_ICLR.cc/2026/Conference — Submitted to ICLR 2026_

### Official Review · Reviewer_ZB3a · 2025-10-30

**Soundness:** 3
**Presentation:** 3
**Contribution:** 2
**Rating:** 6
**Confidence:** 4

**Summary:**

This work proposes a multi-agent framework that automatically generates and aggregates tools into Python libraries. The authors evaluate on three in-domain and one out-of-domain dataset spanning math, medical, and science, and include module-level ablations and an error analysis. The method improves tool-retrieval accuracy, particularly as the tool inventory scales. Consolidating isolated tools into higher-level abstractions is meaningful and using Python library format is a practical choice.

**Strengths:**

1. The core idea—consolidating fragmented tools into higher-level Python libraries—is clear and meaningful.

2. Effectiveness is shown across multiple models and benchmarks (including OOD).

3. Includes an error analysis rather than only headline metrics.

4. The case for “aggregated library is better than fragmented toolset” is convincingly demonstrated under large tool inventories).

5. Ablations are provided for key modules, helping attribute where gains come from.

**Weaknesses:**

1. Retrieval ablations can be deeper. Only one retrieval model is used; no comparisons across alternative embedders/rerankers. No quantitative analysis of how retrieval quality degrades as the tool pool scales.

2. The interaction loop appears single-round (correct me if I am wrong); multi-turn tool use (e.g., Claude-Code) may be more suitable with higher performance and mitigate the limitation of tool retrieval accuracy.

3. Even after aggregation, the resulting tool library remains very large; the degree of consolidation (how much redundancy is removed / how compact the library becomes) is not analyzed.

**Questions:**

1. In many cases, direct code generation may be preferable for advance coding models. How do the authors think of using these generated tools compared with directly generating code?

2. How do you measure the degree of aggregation (consolidation ratio, redundancy removed)? Can the library be further aggregated, and what criteria would govern additional merges?

3. The SFT gain is small; could you break down pre/post SFT by error type and task (what improved, what didn’t)? Which failure modes in your ablation are most amenable to SFT, and which seem unaffected? What type of errors in your ablation study can be effectively solved using SFT?

---

> ### Author Response · Authors · 2025-11-27
>
> **W1**:
> We appreciate the suggestion to deepen the retrieval analysis. We have conducted additional experiments using different retrievers to verify robustness and have analyzed the scaling degradation issue.
> | Embedding model | Fragmented Tools | Tool Library |
> | :--- | :--- | :--- |
> | Qwen3-embedding-8B | 21.7 | 53.1 |
> | OAI Text-Embedding-Small | 24.4 | 58.5 |
>
> As shown in the table (conducted on the 20k question set), our Tool Library consistently outperforms the Fragmented Tools baseline across different embedding models. This confirms that the performance gain stems from the superior structural organization of our library, rather than the choice of a specific retriever.
>
> Cause of Degradation: As the tool pool scales, the fragmented set accumulates a vast number of tools with overlapping semantics but distinct implementations (as visualized in Figure 3, Right). This redundancy creates severe "semantic collision," causing the retriever to retrieve irrelevant or sub-optimal distractors. Our aggregation pipeline explicitly mitigates this degradation by synthesizing these topic-similar tools into a single, comprehensive library.
>
> **W2**:
> We respectfully clarify that there may be a misunderstanding regarding our interaction design. Our framework is already designed as a multi-turn interaction system, not a single-round one. We have revised the paper to make it clear.
>
> **W3**:
> We have quantified the degree of consolidation in Appendix B, which achieves a significant compression ratio, averaging approximately 20 atomic tools condensed into 1 aggregated tool. Despite this high compression rate, the library remains extensive due to the inherent diversity of the large-scale dataset (>100k examples). Besides, we posit that the true measure of a library's quality is not how small it is, but how effective it is in downstream tasks. Therefore, the tool integration accuracy serves as the most appropriate proxy for measuring whether the consolidation was successful.
>
> **Q1**:
> We argue that using generated tools offers superior reliability and efficiency compared to direct code generation. While direct generation is flexible, it remains stochastic and can lead to implementation errors for complex logic. In contrast, our tools are pre-validated assets verified via execution loops during construction, which guarantees functional correctness and stability at inference time. Furthermore, retrieving and invoking a clean tool interface significantly reduces the token consumption and inference burden compared to regenerating complex algorithmic implementations from scratch, making our approach a more robust solution for multi-step reasoning.
>
> **Q2**:
> We control the degree of aggregation primarily through the granularity of our initial tool clustering. The clustering step defines the boundaries for consolidation, while the LLM autonomously determines the specific consolidation ratio within each cluster based on functional compatibility.
>
> Regarding the criteria for additional merges, our stopping mechanism is governed by a critical trade-off between compactness and usability. While further aggregation is theoretically possible, excessive merging risks creating "super-tools" with overly generic descriptions and complex parameter structures, which significantly degrades the LLM's ability to accurately fill arguments. Therefore, we should ensure the library remains compact without sacrificing the granularity required for precise tool invocation.
>
> **Q3**:
> Our qualitative analysis of failure modes pre-/post-SFT reveals that fine-tuning primarily mitigates Parameter Errors (e.g., format mismatches, unit mismatches, etc.). However, Retrieving Errors remained largely unaffected. We attribute this limitation to the current SFT dataset, which consists exclusively of positive instances. Lacking negative samples or contrastive examples to teach the model to distinguish between effective and ineffective queries, the model's ability to refine its retrieval logic remains limited, which explains the modest overall performance gain.

---

> > ### Comment · Reviewer_ZB3a · 2025-11-27
> >
> > Thanks for the explanation and extra experiment. I will raise my score. I like the idea of building tools into a library.

---

### Official Review · Reviewer_nGse · 2025-10-30

**Soundness:** 2
**Presentation:** 2
**Contribution:** 2
**Rating:** 4
**Confidence:** 4

**Summary:**

This paper proposes ToolLibGen, a framework that automatically refactors unstructured collections of tools into structured tool libraries. The method aims to address the retrieval bottleneck that arises when thousands of automatically generated question-specific tools become fragmented and difficult to manage. The framework consists of three major stages, including CoT-based question-specific tool generation from QA data samples, hierarchical tool clustering, and tool aggregation based on a multi-agent coding-reviewing pipeline. Experiments on scientific, mathematical, and medical reasoning tasks and three seed LLMs (GPT-4.1, GPT-oss-20B and Qwen3-8B) show improvements on QA performance over baseline methods including CoT, PoT, fragmented and clustered toolset management, and KTCE.

**Strengths:**

- The research direction of multi-agent tool aggregation and structured tool management is valuable for improving tool usage efficiency when scaling up agentic LLM systems.
- ToolLibGen demonstrates consistent improvements across multiple datasets and foundation LLMs.
- The experimental analysis is comprehensive, where the retrieval scalability curves, error analysis and additional study on supervised fine-tuning framework provide solid diagnostic insight.

**Weaknesses:**

- The creation of tool libraries of ToolLibGen requires a large amount of extra computing cost, and it is hard to keep evolving if new types of problems emerge. It is unclear whether ToolLibGen’s benefits could be naively replaced or surpassed by directly fine-tuning LLMs to create question-specific tools and the solutions with the tools, where instead of retrieving tools from a pre-built fixed library, LLMs are trained to more flexibly create dynamic tools for problem solving at test time.
- The proposed framework requires non-trivial prompt engineering, which limits its generalization to non-programmatic tools without deterministic arithmetic solutions or algorithms, such as search engines.
- The tool library quality created by ToolLibGen heavily relies on the prompted LLMs that conduct the tool clustering and reviewing, whose reliability may be questionable especially on unseen problems.
- The paper writing quality is poor, which includes unpolished duplicated paragraphs (the third and fourth paragraphs in Section 2.4).

**Questions:**

- Is there any efficiency measure to show the computational cost of ToolLibGen at tool library creation and inference phases? It would be better to compare the efficiency of ToolLibGen with baseline prompting and tool-augmented methods.
- In the coding-reviewing tool aggregation step, what is the rate of reaching the maximum number of iterations, meaning that the aggregation may fail? and when this happens, how many tools are still uncovered by the aggregation?
- Any human evaluation to more quantitatively verify the quality of tool creation, clustering and aggregation of ToolLibGen?

---

> ### Author Response · Authors · 2025-11-27
>
> **W1**:
> We appreciate the reviewer's thoughtful comparison between our library-based approach and test-time tool generation via fine-tuning. We address these three aspects below:
>
> 1. Computational Cost: While we acknowledge the computational cost during the offline construction phase, this should be viewed as a one-time investment. In practical deployment, this cost is mainly due to the inference requests. In contrast to test-time tool generation, i.e., requiring the LLM to generate tools for every single query, our approach shifts the heavy lifting to the offline phase. At inference time, our system simply retrieves a validated tool, significantly reducing latency and token consumption.
>
> 2. Evolving with New Problems: Our framework is designed to be extensible. When new types of problems emerge, the extracted question-specific tools can be processed through our existing pipeline. A simple way is to first extract new tools from new questions and then check if they could be merged with the current toolsets or a new aggregated tool is required. While our current experiments focus on validating the effectiveness of the construction pipeline on fixed datasets, the methodology naturally supports incremental updates without rebuilding the library from scratch.
>
>
> 3. Comparison with Fine-tuning and test-time tool generation: While Fine-tuning for test-time dynamic tool generation is a valid research direction, our explicit library approach offers distinct advantages:
> (1) Applicability to Black-box Models: Fine-tuning is often impossible for proprietary models, e.g., GPT-5. Our method acts as a "plug-and-play" module that enhances any LLM.
> (2) Reliability: In the aggregation process, tools in our library are validated offline. In contrast, test-time generation is stochastic; a finetuned LLM might generate correct code once but fail on a similar query next time. Using a fixed, verified library guarantees a higher baseline of safety and stability.
> (3) Inference Cost/Latency: As we discussed before, test-time tool generation will cause an increasing cost and latency.
>
>
> **W2**:
> Non-trivial Prompting Design: We acknowledge that our framework involves prompt engineering, but we respectfully argue that this design does not limit generalization across domains. Our prompts are designed based on universal software engineering principles (e.g., modularity, functional abstraction) rather than domain-specific heuristics (e.g., how to solve a calculus problem). So following this meta-level design, we do not need to carefully design the prompt over time, and the principle itself is universal. We have validated this across diverse datasets, including math, medical, and science QA, and with the in-distribution and out-of-distribution settings. These domains require a mix of rigid arithmetic reasoning and flexible knowledge retrieval. The consistent performance improvements across these distinct fields demonstrate that our designed method is general to different domains, independent of the specific subject matter.
> Applicability to Non-Programmatic Tools (e.g., Search Engines): Even for non-algorithmic tools like Search Engines, the interaction pattern can be structured as a deterministic process. For instance, our pipeline could aggregate specific search actions (e.g., "search for birthday", “search for birth place”) into a generalized tool (e.g., lookup_entity_attribute(name, attribute)). Therefore, as long as a tool can be wrapped in a function interface and the reasoning process is deterministic, our pipeline can effectively work on these scenarios.
>
> **W3**:
> We acknowledge the concern regarding reliance on LLMs for library construction. However, we argue that our pipeline incorporates rigorous safeguards to ensure high quality, even for unseen domains. We address this from three perspectives:
> Execution-based Validation: We do not blindly rely on the LLM's raw output. A critical component of ToolLibGen is the Reviewing Agent (Section 2.4). Any tool aggregated by the code agent is subjected to verification against training queries. Tools that are hallucinated, syntactically incorrect, or functionally invalid fail this step and are strictly filtered out. This ensures that the final tools populating the library are guaranteed to be executable and valid, regardless of the prompt's perfection.
> Task-Agnostic Cognitive Capabilities: The specific prompt provided to the LLM is general and not domain-specific memorization. Therefore, the structural quality of the library remains robust even when the underlying data covers diverse or unseen topics.
> Empirical Evidence on Unseen Data: The ultimate test of library reliability is performance on held-out datasets. Our experiments show consistent performance gains on datasets that were not seen during the tool construction phase. This empirical evidence strongly suggests that our method successfully distills reusable, high-quality tools rather than overfitting to the training data.

---

> > ### Author Response · Authors · 2025-11-27
> >
> > **W4**:
> > We have proofread the paper and removed the redundant paragraphs to ensure the writing meets the conference's standards.
> >
> > **Q1**:
> > We can analyze the efficiency of ToolLibGen using output token consumption as the metric. It is noted that during the aggregation phase, tools are clustered and processed in batches to output the global blueprint and then implement code. This means the cost of aggregation is amortized across all tools within a cluster, significantly reducing the cost per final tool. Besides, this construction cost is incurred only once.
> >
> > For the inference process, we acknowledge that our method introduces additional steps (retrieval query generation and parameter filling in tool calling), resulting in a marginal increase in token cost compared to standard CoT. However, these generated queries and arguments are typically concise, e.g., only one or two query sentences for retrieving, and only some parameters to generate in the tool calling. Therefore, this overhead is not a bottleneck and is a justifiable trade-off for the significant gains in reasoning accuracy and capability.
> >
> > **Q2**:
> > We clarify that reaching the maximum number of iterations does not constitute a "failure," but rather triggers a fallback preservation strategy. In our experiments, reaching the iteration limit is not a very common issue. We sampled 50 clusters and observed that only 6 (12%) triggered this condition.
> >
> > When these max turns are reached, any tools that have not been merged into the current library are not discarded. Instead, they are classified as singleton tools and directly retained in the final library. Consequently, we guarantee all functional retention. Whether a tool is aggregated or kept as a singleton, its functionality is preserved in the library. We have updated the manuscript (Section 2.4) to clarify this mechanism.
> >
> > **Q3**:
> > Evaluating tool clustering and aggregation is inherently subjective for humans, as there is no single "golden taxonomy" for organizing tools. Human agreement on such tasks is notoriously difficult to standardize. Since our library is designed for agents, we argue that functional utility is the most rigorous measure of quality. Specifically, our Retrieval Accuracy (Sec 3.3) and the improved Downstream Performance (Sec 3.2) confirm the correctness and helpfulness of the created tools.

---

### Official Review · Reviewer_vrep · 2025-10-31

**Soundness:** 2
**Presentation:** 2
**Contribution:** 2
**Rating:** 2
**Confidence:** 4

**Summary:**

The paper proposes a method to aggregate tools in agentic large language model systems. The number of tools used by the systems are heavily increasing, and it is becoming a bottleneck for the models to parse through the large collection of tools to use them effectively. This paper proposes a solution by aggregating the tools into semantically coherent clusters. Empirically, this method improves tool retrieval accuracy and overall reasoning performance.

**Strengths:**

The only strength of this paper is their proposed method of using aggregated tools outperforms individual tools.

**Weaknesses:**

- To solve the problem, authors themselves first "invent" the problem by generating a bunch of "question-specific" tools. This makes the problem simpler as authors themselves can define how simple each tool can be, and define an agent which can effectively aggregate the simple tools. For instance, a tool for a Question-CoT pair can be easily hallucinated by the LLM as a copy only tool, and the aggregator can just aggregate a bunch of copy tools and parameterize them. A proper evaluation would have required the authors to use existing tools from a dataset/setup and then attempt to aggregate it.
- The method only works with python functions as tools, as it employs a coding agent which refactors multiple python functions into a single function. This method is not generalizable to tools from other programs.
- The approach uses an LLM to perform clustering of the tools (reaching to a "tool-ception" moment). Interestingly, the authors set the tree depth of the clustering process to 4, thereby making the evaluation favorable to their setup. In Figure 3, authors show retrieval accuracy improves with the aggregation, which is quite obvious as the number of tools to call reduces in the aggregated setup. However, this depends crucially on the tree-depth - higher numbers would likely generate more clusters, leading to less utility of this setup.

**Questions:**

- which model exactly is used for the coding agent, which aggregates the tools?

---

> ### Author Response · Authors · 2025-11-27
>
> **W1**:
> We respectfully acknowledge the reviewer’s concern regarding the tool generation process. We wish to clarify that the "question-specific" nature of the tools is not an artificial constraint we imposed to simplify the problem. On the contrary, our tool generation prompt (detailed in Appendix E) is adapted from prior works and explicitly instructs the LLM to "Make the Python function general." The fact that the generated tools remain question-specific highlights a fundamental limitation in the tool-making paradigm of current LLMs, rather than a manufactured setup. Since manually designing tools for massive datasets is prohibitively labor-intensive, relying on automated generation is essential. Therefore, the inability of LLMs to abstract logic from specific instances is a critical bottleneck that our work aims to solve, rather than a shortcut we created.
> Regarding the "copy tool" hypothesis: One direct evidence against the "triviality" hypothesis is found in our baseline comparisons. If the generated question-specific tools were merely trivial mappings or "copy tools" without substantive logic, using them would provide no benefit over standard CoT. However, our results show that tool-augmented reasoning consistently outperforms CoT. Besides, if the generated tools were merely trivial mappings, our aggregated library would fail to generalize. However, our experiments demonstrate significant performance gains on unseen datasets, proving that our pipeline successfully distills valid, executable logic from these specific instances.
> Regarding the suggestion to use existing toolsets: Manually curated tool libraries are already well-structured and documented by humans. Our research scope specifically targets the automated generation of reasoning tools from CoT data and the organization of these automatically generated tools. Therefore, we believe that showing the effectiveness of a human-created tool is out of our scope.
>
>
> **W2**:
> We chose Python code because it is a standard interface for tool-augmented LLMs and programmatic reasoning in recent literature [1][2]. The core contribution of our work lies in identifying and consolidating common reasoning patterns, which are then organized into an offline Python library to facilitate efficient calls during the inference phase. It is noted that this paradigm for abstracting complex execution logic into reusable modules is not inherently limited to Python. It is a generalizable methodology applicable to any structured environment where reasoning steps can be parsed, merged, and executed.
>
> [1] Gou, Zhibin, et al. Tora: A tool-integrated reasoning agent for mathematical problem solving. ICLR 2024.
>
> [2] Ma et al, Automated Creation of Reusable and Diverse Toolsets for Enhancing LLM Reasoning. AAAI 2025
>
>
> **W3**:
> We agree with the reviewer’s intuition that reducing the number of tools naturally aids retrieval. However, we clearly emphasize that performance gain is not solely a function of count reduction, but rather the quality of aggregation. Aggregation carries a trade-off: merging too aggressively causes functionality loss or parameter complexity (making tools hard to invoke), while retaining too many fails to solve the retrieval bottleneck. The challenge is not just reducing numbers, but reducing numbers while maintaining coverage under reasonable parameter complexity. The fact that retrieval improves and downstream accuracy increases (Section 3.2) proves that our aggregation successfully compressed the search space without losing necessary reasoning capabilities.
> The choice of tree depth is not an arbitrary setting but a design choice driven by semantic hierarchy. A depth of 4 was selected to align with a logical hierarchy: Discipline (Science), Domain (Physics), Sub-domain (Classical Mechanics), Specific Topic (Kinematics). Increasing the depth arbitrarily (e.g., to 10) would indeed generate more micro-clusters, causing the library to devolve back towards the "Fragmented" baseline with high redundancy. Conversely, a shallow depth would force distinct functions to merge. Thus, setting depth to 4 is an optimal trade-off point we identified.
>
> **Q1**:
> GPT-5 is used for the coding agent and aggregates the tool. As we clarify in Lines 145-146 and Line 281-282, GPT-4.1 is only used as a solver LLM during the initial tool creation and the tool review during aggregation, and GPT-5 is used in all other places.  We have indicated the usage of the general LLM and the LLM solver in Fig. 2, but will make it clearer in our revised draft.

---

### Official Review · Reviewer_8vMs · 2025-11-01

**Soundness:** 3
**Presentation:** 3
**Contribution:** 2
**Rating:** 4
**Confidence:** 4

**Summary:**

The paper introduces TOOLLIBGEN, a scalable framework that automatically refactors fragmented, question-specific tools generated from LLM reasoning traces into a structured Python library. It employs hierarchical clustering to group functionally related tools and a multi-agent aggregation system to iteratively merge and validate tools without losing functionality.

**Strengths:**

- The empirical results show the effectiveness in generalization to unseen datasets, which is critical when scaling tool sets in real-world applications

**Weaknesses:**

- Lack of novelty in the core idea or methodology. The core idea of using Python functions as a toolset to aid reasoning is proposed in other previous works [1,2,3]. While the paper successfully showed that having a more generalizable toolset indeed leads to better performance in reasoning tasks, the claim has already been shown in previous papers [3,4,5]. The authors need to show empirical results to distinguish between these works
- Lack of competitive baselines. The only competitive baseline is KTCE [3]. Need more competitive baselines such as [4] and [5].


[1] Qian et al, CREATOR: Tool Creation for Disentangling Abstract and Concrete Reasoning of Large Language Models. EMNLP 2023. \
[2] Yuan et al, CRAFT: Customizing LLMs by Creating and Retrieving from Specialized Toolsets. ICLR 2024. \
[3] Ma et al, Automated Creation of Reusable and Diverse Toolsets for Enhancing LLM Reasoning. AAAI 2025 \
[4] Wang et al, TROVE: Inducing Verifiable and Efficient Toolboxes for Solving Programmatic Tasks. ICML 2024 \
[5] Stengel-Eskin et al, ReGAL: Refactoring Programs to Discover Generalizable Abstractions. ICML 2024

**Questions:**

Address the above questions.

---

> ### Author Response · Authors · 2025-11-27
>
> **W1**:
> We sincerely thank the reviewer for the insightful suggestion to compare our work with prior approaches. We fully recognize the value of these contributions; however, we respectfully clarify that our work addresses a fundamentally different technical challenge: optimizing aggregation within a massive, fragmented toolset, rather than focusing solely on applying these extracted tools in reasoning. While prior methods laid the foundation, they have different focuses: CREATOR and TROVE primarily target question-specific tool generation at test time; CRAFT and ReGAL refactor CoT into code but employ relatively straightforward tool aggregation techniques (e.g., deduplication); and KTCE only relies on pairwise tool mutation without a global strategy. In contrast, we introduce a clustering process to decide groups of tools to aggregate and an aggregation process to make a global design and generate new, general tools. Moreover, we represent a step forward in scalability (>100k examples) compared to previous efforts (~5k), validating the potential of scaling up this research track.
>
> **W2**:
> We appreciate the reviewer’s suggestion regarding competitive baselines. We would like to clarify that our baseline selection was strict regarding the specific goal of our framework: improving aggregation process of question-specific tools. Consequently, we did not include TROVE as a direct baseline because it focuses on test-time generation rather than offline construction. Similarly, while ReGAL is a valuable work, its organization strategy is primarily only based on deduplication. We established KTCE as our primary baseline because, to our knowledge, it is the state-of-the-art method specifically targeting tool compaction. Thus, KTCE represents the most rigorous benchmark to demonstrate the effectiveness of our novel pipeline.

---

### Meta-Review · Area_Chair_E459 · 2026-01-03

**Summary:**

The following four types of concerns can be summarized below:

(1) Novelty Limitations. Reviewers questioned the novelty and generality of the core methodology. The core idea of using Python functions as tools for reasoning was deemed lacking in novelty compared to prior works. The framework is restricted to Python functions, raising doubts about its applicability to tools from other programming languages or non-programmatic tools. While tool aggregation is a core component, it may be perceived as a marginal innovation rather than a primary contribution.

(2) Generalization Concern. Concerns were raised about the artificial nature of the "question-specific" tools generated by the authors, as they may simplify the aggregation problem. The framework’s reliance on prompted LLMs for clustering and reviewing raises questions about its reliability for unseen problems. Additionally, the clustering process relies on LLMs with a fixed tree depth of 4, which reviewers considered a setup favoring the method rather than a universally valid design.

(3) Efficiency Concern. The tool library requires high computational costs for construction and faces challenges in evolving with new types of problems, while the need for non-trivial prompt engineering may limit its adaptability. Furthermore, the paper lacks sufficient efficiency analyses, including end-to-end computational cost and latency comparisons with baseline methods during both tool library construction and inference.

(4) Unconvincing Experimental Setups. The choice of baselines is limited, with only KTCE included as a direct competitor, excluding other relevant methods like TROVE and ReGAL. Experiments rely on self-generated tools rather than existing real-world tool datasets, which reviewers argue is not a proper evaluation. The paper lacks human evaluation to verify the quality of tool creation, clustering, and aggregation, and the use of functional utility metrics alone is considered insufficient.

**Reviewer Concerns:**

The rebuttal may address the reviewer's concerns:
- For unconvincing experimental setups, the authors supplemented retrieval experiments with multiple embedding models, confirming the library’s structural advantage.

- For efficiency concerns, they quantified the consolidation ratio and explained that construction costs are one-time, while inference overhead is minimal. They also clarified the iteration limit mechanism, noting that unmerged tools are retained as singletons to ensure full functionality.

The reviewer's concerns may still be outstanding:

- The core methodology’s novelty remains questionable, as the distinction from prior works’ aggregation techniques may not be sufficiently impactful. The framework’s generalization to non-Python and non-programmatic tools lacks empirical validation, despite theoretical claims.

- Reliance on LLMs for clustering and reviewing still poses risks, as execution-based validation may not fully mitigate hallucination or domain adaptation issues. The computational cost of library construction remains a practical barrier for large-scale deployment, and the incremental update mechanism lacks detailed experimental support.

- Efficiency analyses are still insufficient, as token consumption metrics do not fully reflect real-world latency and hardware-level costs. The consolidation ratio, while quantified, does not address whether further aggregation is feasible or beneficial. Retrieval analyses, though supplemented with additional embedding models, still lack exploration of scaling degradation mechanisms and comparisons with advanced retrievers/rerankers.

- The baseline selection remains limited, as excluding TROVE and ReGAL may understate the competitive landscape. The use of self-generated tools instead of real-world datasets still weakens the evaluation’s external validity. Human evaluation is still absent, and functional utility metrics alone may not capture the quality of tool organization for human or agent use.

**Reviewer Scores:**

- For Reviewer 8vMs, he may not change the score, as concerns about novelty and additional baselines may persist.
- For Reviewer vrep, he may not change the score, as core concerns about artificial tool generation, Python-only limitation, and the clustering setup remain insufficiently resolved.
- For Reviewer nGse, he may not change the score, as concerns about computational efficiency, LLM reliability on unseen data, and lack of human evaluation are still not fully addressed.
- For Reviewer ZB3a, he may raised the score due to the authors’ supplementary retrieval experiments and clarifications on multi-turn interaction and consolidation ratio. However, some concerns about retrieval depth and SFT failure mode analysis may remain.

---

### Decision · Program_Chairs · 2026-01-26

Reject